# The Spaces and Places of the Tourism Encounter.
# On Re-Centring the Human in a More-Than/Non-Human World

Edward Hákon Huijbens

Cultural Geography, Wageningen University & Research, 6708 PB Wageningen, The Netherlands;
edward.huijbens@wur.nl

**Abstract:** This paper will revalue the phenomenological understandings of the tourism encounter, inspired by spatial theories of intentionality. With a growing body of theory delving into the relational realm and the ways in which the body and our actions are relationally enmeshed in networks of more-than/non-human entities, this paper seeks to recentre human intentionality as the core of the tourism encounter to better address its political nature and relevance. Whilst thereby critiquing some of the propositions of relational ontology, the paper is not about rejecting these, but augmenting them through a focus on the intention to care. Thereby, the paper will explore the ways in which the tourism encounter can be re-storied as one for making spaces and places of conviviality through people relating to each other and their surroundings with particular intent imbued with care. Valuing care and how it can be narrated helps to make space for a plurality of futures which can in turn break the deadlock of tourism being conceived either as mass/over- or alternative tourism. Both of these and more exist at the same time in the same place.

**Keywords:** encounter; conviviality; intentionality; tourism; care

## 1. Introduction

The phenomenological enquiry pivots on understanding the ways in which the subject relates to the object world, posing the question what is 'is' in terms of the things that are made sense of. This enquiry thereby simultaneously delves into both the material, object world and the subject world through the ways in which they are connected and can be known. One of the central concepts in making sense of this relation is 'intentionality', which refers to the directedness of consciousness towards objects. According to the founding figure of phenomenology Edmund Husserl, all conscious acts have intentionality; they are always directed towards something. For example, when we see an object, our consciousness is directed towards that object, and we perceive it in a particular way.

However, this object world is by now most commonly referred to as the more-than/non-human world in scholarly attempts at broadening agency. Thereby, sense is made of how, for instance, non-human animals and the material also have ways of relating and thereby have a role to play in making spaces and places (see, e.g., Bennett 2010). As agency is further assigned to the more-than/non-human, studies have also delved further into the relating body and what it can do in its proprioceptive and sensuous capacity (Massumi 2015; Simpson 2021). The basis of phenomenology rests on an analysis of lived experience and the fact that people think with their whole bodies. Explorations of sensuous geographies and extending concepts of intention and reason beyond the humanist subject add analytical perspectives, and through these explorations, recent phenomenology has helped to do away with a particular kind of human, i.e., the human who thought himself (indeed usually white and male) able to perform abstracted reasoning, detached from their own body, sensory apparatus and the environment in which they are embedded. In this sense, phenomenology has contributed to notions of the post-human (Braidotti 2013), or revealing the human as a relating desiring machine, to pick up on the wording of Deleuze and Guattari (1988), whereby there is continually an 'and' to our being, not either/or.

This diffusion of agency, however, has critical implications for how the "knowing subject" is assembled. The post-human desiring machine is emblematic of our 'post-representational times' (Day 2005), wherein what mediates habitual and practical existence has become obscured as simply a 'generative middle ground'. Such 'passive vitalism' (Colebrook 2010, pp. 80–81) and 'post-phenomenology' (Ash and Simpson 2016) represent a fundamental challenge in retaining the power of the subject to do important critical work, whilst recognising its post-human state. Thus, these relational approaches have come under a variety of critiques, which range from those wanting to reinstate the (post- or more-than-) human at the centre (Hepach 2021), to harkening, to metaphysics (Kinkaid 2020). Following the former, this paper will challenge the hegemony of relations, reinstating human intentionality, in particular through the intention to care. Intention is a meaningful and important part of our ever ongoing relationality and becoming, yet it is indeed haunted by the spectres of non-representation.

Being acutely aware of what has become planned obsolescence of theory in academia, symptomatic of our current 'accelerationist times that generate endless new 'turns' and new 'fields" (Dillet 2017, p. 519), I aspire to redress phenomenology around the human in the context of the tourism encounter, mainly drawing upon explorations in geography and studies of places and spaces. The point of departure is a sympathetic critique of relational ontologies, yet wanting to add a dose of realism to the mix along with political relevance. The former is a particular kind of realism inspired by Shaviro (2014). He claims that the phenomenological assumption rests on perception and sentience as fundamentally and necessarily intentional. Inherently, phenomenology is thereby about phenomenal appearances and thereby profoundly correlational and fundamentally people-centred. To come to terms with this, Shaviro (2014) proposes 'speculative realism', drawing attention to the human as being creative and the role of aesthetics in comprehending the world. Similarly, de la Bellacasa (2017) calls for situated speculative ethics to account for how we choose to enact our care and for what. She explicitly draws on the work of Kaufman-Osborn et al. (1993), whilst she 'gently' decentres the human in order to prolong relational ontology's ongoing problematisation of any claims. de la Bellacasa (2017) is, however, focusing on soils to broaden perspectives "without discharging humans from specific and situated ethico-political response-abilities" (p. 217). For me, re-centring the human without discharging the more-than/non-human is of relevance when redressing the phenomenology of the tourism encounter. Thereby, the political relevance lies in grasping the transformative promise often associated with the tourism encounter (Pritchard et al. 2011) through the intentions articulated. Moreover, this re-centring helps in making sense of the challenges that humanity, albeit highly differentiated, faces in times of climate change, as well as the hegemony of globalised capitalist consumption peddling individual interests and desires and deliberately dismantling collective welfare for the sake of privatised profit. Indeed, strong trends can be observed towards individualism and the foregrounding of individualised emotion in recent decades at the expense of collectivism and rationalism (Scheffer et al. 2021). Whilst coming to terms with our emotions and embodied states, we need to prevent this from degenerating into individualism appeased through consumption. Through reinvigorating that which we attach meaning to and mediates our collective actions, wary of reducing this process to pure relationality and difference along with narcissistic self-centredness, the tourism encounter gains salience as a platform of enacting intentions. In a nutshell, I will reassert the human in the 'generative middle ground' of relational ontology through intentions of collectivism, mutual aid and care for the other in a broad sense. In this, inspiration is drawn from the explorations of Graeber and Wengrow (2021), whereby they explain history from the perspective of care, challenging the following:

> If mutual aid, social co-operation, civic activism, hospitality or simply caring for others are the kind of things that really go to make civilizations, then this true history of civilization is only just starting to be written. (p. 432)

This paper has been developed in three parts, beyond the introduction and conclusion. First, it will outline a spatial take on phenomenology, drawing on geographies inspired by the existential and later hermeneutic phenomenology of Martin Heidegger translated into the work of Bourdieu and Lefebvre. Thereafter, the paper will show how these have developed into the more-than/non-human relational theorisation, roughly summed under the label 'post-phenomenological' approaches. Lastly, the paper will focus on the tourism encounter to reiterate the roles of caring and hospitality as the intentions centring the human in spaces, to be conceived as convivial.

## 2. Place Ballets

Everything that is experienced has a location in space, is open to be experienced and is set at a distance from ourselves and others. This geography of phenomenology has by now a legacy extending to the 1970s. Tuan (1971) plainly states that phenomenology to him meant the following:

> ... a philosophical perspective, one which suspends, in so far as this is possible, the presuppositions and method of official science in order to describe the world as the world of intentionality and meaning. (p. 181)

To Tuan, phenomenology is about how meaning is made of the spaces that we encounter and experience. Doing geography and understanding what space and things in themselves mean to the intentional subject then reveals the human, and thereby, to know the world is to know oneself (Relph 1981). Delving into this meaning making, Seamon (1980) brings attention to what phenomenology calls the natural attitude, i.e., the unnoticed and unquestioned acceptance of the things and experiences of an individual's lifeworld. Implicit is the revelation of the preconscious processes that guide behaviours. According to Seamon (1980), this is an inherent capacity of the body and thus it can be conceived of as a 'body-subject'. The body-subject then makes for a body-ballet, which is a set of integrated behaviours which sustain a particular task or aim. Seamon (1980) sees this dancing metaphor as intricately linked with identity. She goes on to dispel the link made between identity and community or territoriality, as she claims that they are based on a "post-mortem autopsy—'grieving for a lost home'" (p. 169), similar to MacCannell's ([1976] 1999) discussion around authenticity at the same time. Beyond the focus on embodiment, these early phenomenological takes show how training and practice make the fusion of the movement of the body-subject into a preconscious choreographed performance of flow and rhythm, which implies the organic and integrated nature of the body-ballet. The multiple body-ballets and emerging time–space routines are conceived of as habitual bodily behaviours, fusing and forming a pulsating place ballet. This means that space is first and foremost grounded in the body, the lived body, be it preconscious or not.

These ruminations were built, according to Pickles (1985), on a particular Husserlian version of phenomenology, which he deemed to be inadequate for the study of spatialities. In turn, he advocated for the Heideggerian existential revaluing of Husserl as an approach to remedy a fundamentally unreflective enquiry that he claimed characterised earlier work, with its focus on the body-subject. Heidegger's early phenomenology was dealing with the ways in which the world makes itself available to the knowing person, recognising that "in order for something to be something, it must first be. Being in general is the condition of possibility for being in particular" (Elden 2001, pp. 9, 22). Thus, Heidegger's concern was the distinction between being and Being, or the distinction he made between ontic (examining the nature of being through observation and empiricism) and ontological (what is the condition of possibility for the ontic) knowledge. The Heideggerian view is that temporality is an integral part of the human experience, as the present draws behind itself a 'comet's tail of retentions' in the Husserlian sense. Flaherty (1999) argues that humans are unique in the sense that experiences of heterogeneous events are fused into a coherent sense of persistence. Change is managed by remembering the past, stepping back from the present and anticipating the future. Thereby, Flaherty (1999) opens up the question on

how the passage of time is perceived from one situation to another in social reality, making room for intention.

With focus on intention, Heidegger called upon 'Dasein' as a being whose Being is an issue for the here and now (being-there). This is "that entity in its being which we know as human life; this entity in the specificity of its being, the entity we each ourselves are, which each of us finds in the fundamental assertion: I am" (Collins and Selina 1999, p. 51). In Heidegger's philosophy, Dasein's ontological basis is in experience, and it precedes all knowledge, or 'existence precedes essence', as the Sartian dictum goes, itself based on a reading of early Heidegger (Dreyfus 1991). Heidegger analysed Dasein in terms of its concerning being-in-the-world, manifesting in two modes of concerns. The first is related to beings that are ready-to-hand for Dasein, i.e., ready for practical use, with ascribed meaning, functions and significance that have been construed by those ready-to-hand beings in relation to other beings, or themselves fundamentally being-in-the-world. The second is related to beings that are present-at-hand, i.e., those of not immediate practical use, but seen through detached observation, abstraction or contemplation, e.g., the wind or the current of the sea. This presence-at-hand is not of immediate use to Dasein as is the ready-to-hand, and hence for Dasein's concerning being-in-the-world, the concern regarding that which is ready-to-hand has primacy. Thus, in order to understand Dasein's Being, it must be examined in its practical everydayness. Or, in a word, Dasein's being-in-the-world comes through its skilful engagement with objects in the world, more specifically, being-there, in a complex context of significance mediated by concern (Polt 1999). It is this Dasein as being-in-the-world of beings ready-to-hand that is the object of concern when it comes to the tourism encounter and placing intentionality.

*Placing the Encounter*

Comprehending subjectivity as Dasein, or being-there with concern for that which is around you has been operationalised for enquiry in the work of Bourdieu and Lefebvre when it comes to grasping the dynamics of an encounter and the making of spaces and places. Heidegger was a mutual influence on both Bourdieu and Lefebvre. Lefebvre's explicit debt extends to The Production of Space, where the notion of 'lived space' holds resonance with Dasein's being-in-the-world (see Lefebvre 1991, p. 121), as does Bourdieu's embodied notion of habitus (Elden 2004, p. 98).

By turning to the body, Bourdieu explains practical sense, specific to the place in which the subject acts through the body at each moment. Through the use of words like skills, competence and dispositions, the focus is on what the body of the subject does through what Bourdieu termed habitus, also indicated by the simple origin of the concept of the verb 'habit'. The subject, through the mediation of the habitus, is decentred and socialised. Schinkel (2003) adds that the notion of the body as habitus is especially effective, as it works on levels below the reach of introspective thought and that of will. Bourdieu (1998) claims that cognitive reckoning with the world is an objective structure already applied to the dispositions of the body. Hence, what attaches us to objective structures is a tacit or immediate prereflexive agreement between the objective and incorporated structures. This prereflexive agreement relies on the habitus to be understood as layers of embodied experiences and not immediately open to self-fashioning, for example, mediation through the rational action of a reflexive subject (Bourdieu 1998). To Bourdieu, self-fashioning can only take place through practical mimesis, i.e., a kind of subconscious imitation of objective structures and action unfolding around the subject; thus, habitus is dynamic but lacking in intentionality (McNay 1999).

As stated, for Bourdieu (1998), cognitive reckoning is already an acquired disposition. But, Bourdieu does not elaborate on the ways in which the incorporation of objective structures and practical mimesis take and make a place. Bourdieu (1990) merely states the following:

the habitus is an infinite capacity for generating products-thoughts, perceptions, expressions and actions–whose limits are set by the historically and socially

situated conditions of its production, the conditioned and conditional freedom it provides is as remote from creation of unpredictable novelty as it is from simple mechanical reproduction of the original conditioning. (p. 55)

As the habitus refers to one's sense of place and role in the lived environment, and as it works primarily below the cognitive, then all reactions to a situation are an intuitive practical reaction based on experience. This precognitive level of intuitive practical action develops over time through the incremental ingraining of experience, and thus is habitus. Drawing on Merleau-Ponty's (1962) 'habitual body memory', albeit not explicitly (see Bech 2021), habitus is thus about the slow, cumulative 'sedimentation' through repetition, which builds the depth of the body's often unconscious experience of its milieu and of its own possibilities of action. By basing the concept in precognitive experience, the only way the habitus is open to change is through the encountering of other different ways of doing, i.e., other forms of habitus, as there is no specific way of determining change from within or any kind of plasticity to the notion itself (Painter 2000). This leads to the habitus' determination of the subject's actions; at least, it is difficult to see how individual actors can transcend this particular frame of preconscious mediation (Noble and Watkins 2003). But, tying it with Lefebvre's notion of lived space gives the relation between objective structure, in this case the physical material space, and the body some salience and cause for explicit intent.

For Bourdieu, the general social world is where concerning relations and practical dispositions play out politically, thus fleshing out what Heidegger did not, i.e., what people's subjective concerns are. But, the same applies for physical material space that Lefebvre argued for in terms of lived space, drawing inspiration from Dasein. Lived space is one of the concerns prior to any abstraction or contemplation, the space of experience which can stretch and distort in lieu with concern. Space is thus opened up, in one sense, by the concerning presence of Dasein enacted through the embodied habitus. What this opening up entails is not a contemplation of the world by Dasein, but an unveiling or disclosure of the world in a particular way. In short, Dasein's concerning way of dealing with the world discloses a pragmatic spatiality, productive of differentiated spaces, be it formalised events, serendipitous encounters or simply being there and then. As a consequence, Dasein's pragmatic spatiality must be ontologically primary in order to understand everyday spatiality. Heidegger (1971) explained the disclosing of pragmatic spatialities in terms of dwelling, which has since been taken up when grasping the nature of being and framing the ways of relating (Ingold 2000).

In the context of the tourism encounter, caring and hospitality can become the practical schemes and conduits of intent. Following Cooper (2014), the tourism encounter, enacted as pragmatic spatiality, can inform 'everyday utopias' in moving "away from notions of flawless static ideals to a concern with process, change and conflict" (p. 127). Our habitual everyday practices can be changed through the tourism encounter if it is centred on care and hospitality. To which degree these everyday utopias emerge purely through the process of relating and being there is at the heart of this paper. As such, it queries what is gained and lost by reclaiming intentionality in our post-representational times and where we can construct a meaningful threshold for intention in the tourism encounter.

## 3. Relating

The ways in which Heidegger described tools as being ready-to-hand "gives them a strange autonomy and vitality" (Shaviro 2014, p. 48), prompting an exploration into the agency of things in themselves and how "objects are irreducible to simple presence" (Shaviro 2014, p. 51). Through these object orientations (Harman 2018) and later engagements with notions of dwelling, Ash and Simpson (2016) discern a particular development in spatial engagements with phenomenology, which they label 'post-phenomenology'. They freely admit that the label is rather loosely operational, but claim that three particular emphases differentiate this particular body of theory from that of the phenomenological. Therein, intentionality is conceived as an emergent relation with the world. Secondly, objects are understood as having an autonomous existence outside of the ways in which

they appear to be or are used by human beings. Lastly, they discuss placing our irreducible being with the world as being central to enquiry, i.e., there is no above or beyond, only with, and we can only see the world horizontally through the unfolding relational mesh. It is here that my own work can very well be placed (see Huijbens 2021a), as through my studies I was inspired and directly supervised by some of the key figures that Ash and Simpson (2016) identify as the proponents of post-phenomenology (see Anderson and Harrison 2011). Therein, problematics around the body, practice and performativity are reassessed through ideas of non-representation and how to engage with the taking-place of everyday life (Simpson 2021). Drawing on a particular Deleuzian-process-oriented take on phenomenology, the productive potential of relations in terms of immanence and emergence is introduced. This is premised upon the exteriority and irreducibility of relations to their terms, or stating that it is basically through the relating that relations are inherently productive in themselves.

To contrast this, recall Bourdieu, claiming that every day we act and do things that are not that readily explainable but at the same time sediment in bodily dispositions, and as Lefebvre added, spatial formations that structure further encounters. According to 'post-phenomenology', these structuring dispositions and spatial formations are not settled, but driven by their inherent ceaselessness and so continually press on. As a consequence, they can never be fully determined and allowed a singular voice in the struggle for differing aspirations, but need to be thought of in terms of their potentiality. Therefore, in terms of the tourism encounter, they are multiple and innumerable, and cannot be categorised in definite terms. Whilst this way of explicating how we relate is true, in the tourism encounter, it might be pertinent to adopt a type of 'strategic anthropomorphism' of these relations (Caracciolo 2021, p. 160; citing Malafouris 2013; see also Shaviro 2014, p. 61).

*Beyond Relation*

In our everyday life, we do categorise, and we do take cognitive stands based on wishes, desires, wants and aspirations—our intentions and concerns. At the same time, place is full, sensual and immersive, representing the habituation of bodies and presence (Kingsbury and Secor 2020, p. 11). Herein lies the challenge of adding an intentional body-subject to the 'post-phenomenological' mix of relational emergence. To address this challenge and allow a 'strategic anthropomorphism' to take shape, an analysis of the ways in which we do our 'philosophizing, storytelling and art-making function as inevitable technical prostheses for a human engaged in the theorization of matter' (Zylinska 2014, p. 136) is necessary. In other words, how we speculate about that which we call 'real' (Shaviro 2014) and how we tend to that (de la Bellacasa 2017) matters. de la Bellacasa's (2017) speculative ethics invite us to engage in imaginative and reflective thinking to consider the ethical implications of future possibilities. Thereby, we are called upon to think critically about the values and principles that underpin our decisions and actions. The tool to come to terms with speculation and make space for the intentional body-subject is storytelling, and the stories that we tell and choose to attach to:

> . . . it is by cultivating an imagination of abstract pattern that narrative can move beyond isomorphic projection, or naïvely anthropomorphic accounts of the nonhuman. (Caracciolo 2021, p. 163)

'Narrating the Mesh' is thus more than just seeing and telling; it is about creatively navigating the abysmal chasm that the post-phenomenologists make so much of, and is evident in the following question: what is 'is'? As Olsson (2007) claims, to be believed is to have power, power is the desire to control meaning and the prime symbol of meaning is the copula 'is', a verb designating an event taking place in the void of the excluded third. In Olsson's (2007) terms, we need to recognise the limits of language as being at the cusp of this void, which can only be navigated by naming. According to Olsson (1998), these are as follows:

the combined principles of geometry and naming, i.e., in the interface between the theories of picture-making and story-telling, on the one hand, and the practices of pointing and baptising, on the other. (p. 147)

Telling the story and creating the image at the same time that one points to things is how meaning is made to matter and places get animated (Rose 2020). Neat as it is, it does also indicate the indirect causation of attunement and deep time, and Kotva (2019) is pointing us towards this with reference to Heidegger in the context of climate change.

The concept of habit as a kind of tuning is vital to the Anthropocene because it shows how consciousness can make things happen, even when it looks as though it is having no direct impact on the planet. (Kotva 2019, p. 243)

Thus, also in the grandest of contexts, the planetary, places and how we enact them matter. But, this link also hints at its impossibility, paving the way for ongoing inconclusivity and hence the open-endedness of our being, which then generate potential and beginnings, not closure from the event of being here and now. As Olsson (2020) concludes the preface to his latest work:

Breathe normally. And you too might experience how the prow shears through the night and into the dawn. (p. xi)

As breathing itself has a distinct political geography (Pratt 2022, p. 278), Olsson is moving beyond the extensiveness of our being, our ongoing openness, and draws attention to how we make meaning matter through exercising the power of naming in the simplest of everyday acts. Here, intention emerges, beyond human exceptionalism, anthropocentrism and transcendentalism, wherein questions of which human is reified become pertinent with ramifications for tourist studies (Cohen 2019). The in-between being allows for the subject through the practices of naming and telling a story, as well as being believed in doing so.

## 4. The Tourism Encounter

The tourism context in which subjects relate to the object world has been explored for some time. It dates back to Simmel's ground-breaking work on urban strangerhood as a phenomenal characteristic of modernity (see Wolff 1950) throughout the early 70s, where efforts were being made to place tourism as a sub disciplinary domain within anthropology, understanding social–cultural impacts and stressors that needed to be understood in earnest (Smith 1978). Whilst more squarely focused on how "tourism can be a bridge to an appreciation of cultural relativity and international understanding" (Smith 1978, p. 6), later foci were on the tourist's experience and tourist types (see Cohen 1996; Bauman 1996). In the 1970s, MacCannell ([1976] 1999) discussed the tourist experience as a form of social performance in his revisiting of Veblen's classic on theorising the leisure class, and thereby how spaces and places can be managed and manipulated in making the tourism encounter. The more explicitly phenomenological Crouch et al. (2001) later explored how tourists engage with and interpret their travel experiences, highlighting the importance of understanding tourism as a deeply personal and subjective phenomenon, rather than solely as an economic or social activity. They emphasised seeing the tourist as agentive and embodying spaces and places through encountering these. Whilst such promising indications exist, Pernecky and Jamal (2010) claim that attempts of engaging tourism experiences with phenomenology have inadequately addressed the theoretical and philosophical assumptions that influence the research. Indeed, a lot of work dealing with tourism experiences in the post-2000s is caught up in the Experience Economy formulation of Pine and Gilmore (1999). To remedy this, Pernecky and Jamal (2010) offer an engagement with Husserl and Heidegger in outlining a hermeneutic epistemology grounded in realist ontology. Trying to address the limitations identified by Pernecky and Jamal (2010), Fendt et al. (2014) embrace Heideggerian phenomenology to investigate the experiences of being a surfer. They challenge the researchers of tourists and tourism to adopt vivid writing and visual aids to allow for immersion in the experience being recounted.

Indeed, using phenomenology, rich and candid understandings of tourism experiences can be acquired, opening up the encounter with individual creativity and that of the researcher. Crouch's (2010) 'flirting with space' similarly sees space as in constant flux, lending itself to our creative acumen and highlighting the subjective nature of tourism encounters and the 'gentle politics' involved. Thereby, through Crouch, tourists' lived experiences, the transformative nature of tourism and its impact on individuals and societies can be gauged. Partly inspired by this, I can place my own work (Huijbens and Benediktsson 2013), where we argue that the tourism encounter is simultaneously an effect of gathering deep-seated emotions and experiences and an open-ended and forever unfinished story taking place. Similarly, Prince (2018) infuses the tourist landscapes with the non-representational ethos of materiality and embodiment in provoking a conversation on the complex, yet mundane, experience of inhabiting tourist landscapes.

In sum, studies of the tourism encounter have sought to understand the ways in which the "'tourists' of modernity morphed into a human subject enunciating pleasure, hedonism, adventure, and educational thirst (or greed)" and studies "which center on the problématique of perception and performance" (Tzanelli and Korstanje 2020, p. 61). Tourism as performed in place is where the debate around the phenomenology of tourism and travel can be placed, and it is indeed within this problématique that we need to stay (Ren 2021) and wherein tourism is enacted.

The above overview of phenomenological tourism explorations makes clear that place matters. The open and emergent sense of the human, as proposed above, sets the world as neither a subject nor an object of representation. Thereby, one must enter into a more active 'sense' of the world. But, this sensing ultimately revolves around spacing (Doering and Zhang 2018), and the taking and making of place. Studies in the urban already give indications of the role of encountering the material in spacing and placing. Tonkiss (2013, p. 313) interrogates the critical spatial practices that operate "in the cracks between formal planning, speculative investment and local possibilities", and as Barba-Lata and Duineveld (2019) argue, the focus can therefore be on "minor practices, small acts, ordinary audacities and little anti-utopias that nevertheless create material spaces of hope in the city" (p. 323). Barba-Lata and Duineveld (2019) build on Tonkiss (2013) as they explore what they call 'the productive potentialities often hidden within the materialities of the urban', citing Latham and McCormack (2004, p. 719). Recognising these and "the ways these become a vivid source for an alternative value regime and related narratives of belonging is merely a first step" (Barba-Lata and Duineveld 2019, p. 1771) for a more critical and imaginative mode of being.

### 4.1. Recentring the Human

Following Heidegger, Dasein's pragmatic spatiality must be ontologically primary and the object of concern when it comes to the tourism encounter and ways of understanding intentionality. Being-there with intent quite literally matters. These intents may have sedimented into bodily dispositions through force of habit, and are thereby not discernible as outlined intent, but are there nonetheless. Drawing in this way on the phenomenology of Heidegger and later Bourdieu and Lefebvre, the very place (the 'there' of being) matters, and through relating to the materiality of place, spatial structures of various kinds emerge and make the meaning making that in turn matters again. At the same time, our intentions in relating play a role.

In the process of setting up the distance between us and the other, people can indeed be seen as relational beings and agency is assigned to all things material and immaterial. Thereby, people emerge as nodes in the endless actor-network of being (Van der Duim et al. 2012). Following practices and doings in making these relationalities and looking at the world sideways along these relations is a fun thought exercise, but at the end of the day, it is us, through our deliberations, the stories we tell, the myths we create and the meaning that we make, who are making the places as they make us. Büscher (2021) reminds us that

acknowledging certain forms of human exceptionalism is no impediment to recognising more-than-human relations, regardless of what he labels 'the non-human turn':

> ... we must place ourselves firmly within the dialectical tension that the co-constitution of nature and society represents, which is always an epistemological, analytical, and political balancing act that responds to forces of power and other (inter)relationships. (p. 5)

In other words, precisely because we make the places which in turn make us, we need to distinguish between the different elements at play to meaningfully understand the relations that constitute their inter-relation. The spatial interrogation of the tourism encounter is thereby one of understanding the " ... routine calculus of exchange with unknown others: ongoing negotiations of proximity and boundaries premised on varying degrees of familiarity, intimacy and trust" (Koch and Miles 2021, p. 1380). Place, as the material condition informing the routine calculus of exchange, governs both the subject and object and " ... guarantees the coherency of subject and object in experience while allowing them to be set apart" (Hepach 2021, p. 12).

Through travel and the tourism encounter, the negotiations of proximity and boundaries are paramount in making the encounter and these most profoundly rest on our ability to care. In the American indigenous tradition of braiding sweetgrass, Robyn Wall-Kimmerer calls for the following:

> ... our responsibilities as human people to find ways to enter into reciprocity with the more-than-human world. We can do it through gratitude, through ceremony, through land stewardship, science, art, and in everyday acts of practical reverence. (Kimmerer 2013, p. 190)

Kimmerer's weaving of Indigenous and Western knowledge systems in proposing a more ecological worldview and degree of humility to our ongoing concerns is very much in line with the ways in which one can conceive regenerating tourism (Bellato et al. 2023). But, the reverence and responsibility called for appeal not only to the more-than/non-human world, but also to that of other people. When it comes to tourism, our ideas of the world, intentions and actions of spacing matter, but they do so resting on the "gift and surprise that is the Other can only wander in when that space is open" (Farage 2013, p. 46).

### 4.2. Reclaiming the Tourism Encounter

Foreclosing tourism and destinations as spectacular experiences to be ticked off the to-do list or harkening to naïve notions of personal transformation is not only a closure of space, but also of the mind. The tourism encounter needs to recognise us as spawned by the relations forged through encounters with others and the more-than/non-human, but animated by intent and care which we explicitly need to come to terms with. Here, the image of tourism can be illustrated with notions of conviviality:

> ... rest[ing] immensely on the imaginative potential of the individual and the everyday use of tools rather than premised on a political mass mobilization or institutional territories of anti-industrial resistance. It is through the individual rediscovery of everyday life and tools, we begin to imagine convivial commonwealth alternatives to industrialism, cultivated and vitalized as social challenges to industrial forms of life. (Atasay 2013, p. 63; citing Illich and Lang 1973)

Caring about each other and the places we visit and encounter can indeed challenge the industrial forms of tourism which manifest as mass or over-tourism, or are marketed notions of tourism's potential. Staying true to the post-phenomenological tradition, Wise and Noble (2016) identify the productive possibilities of conviviality as residing in the potential ambivalence at the heart of our embodied, affective and sensory interactions with each other and the material environment. This possibility, they state, emerges under the terms of ever-increasing diversity and diversification, growing mobilities and growing experiences of radical difference. This calls for attention to belonging as practice, whereby

what mediates this 'with-ness' plays a role, and it is imperative to empirically examine the situated practices and performances making conviviality.

Whilst recognising these concerns as animating the enactment of tourism and the tourist encounter, I have previously proposed exploring conviviality through the very stories that rocks and steam afford us (Huijbens 2021c). How these can create spaces of conviviality is through the stories told of these encounters about other ways of being and doing. Here, mundane everyday materialities can become a spectacular experience that we all have access to, even in our backyard, and travel is not only about transporting our bodies to places mediated to us as spectacular experiences, but can also be about flights of fancy. The stories we tell and make matter, making spaces of conviviality wherein alternative world views can be forged, and they gain hold, informing all of the myriad ways in which life can be reconceived, together with the places and spaces that we hold dear. I have explored this in the context of the city of Amsterdam (Huijbens 2022), where I conclude with finding the tourism encounter to be about 'radical hospitality', focused on the embodied lived spatial practices of the everyday in coming to terms with urban strangerhood (see also Simonsen and Koefoed 2020). What arguably emerges is the multiplicity of practices that make up the place of the tourism encounter. By giving in, the art of paying attention to the possible can be cultivated, precipitating an experimental form of everydayness that relies on creative energies and desires, whereby people are not consuming individuals, but individuated parts of the destination as it is made and remade every day through varying degrees of intent. We are indeed relational beings and we need to allow for this, but with intent.

> Staying within an open and emergent sense of the world forefronts the "practical" act of patient, considered analysis as a basis of respect in a singular plural world. (Doering and Zhang 2018, p. 234)

'Staying with the trouble', as Haraway stated (cited in de la Bellacasa 2017), is about being open to being relational, but comes along with the responsibility of the forcefulness of our being. The stories that we tell and how we make sense of the world most certainly matter and need to be taken seriously in understanding the tourism encounter. In the stories that we tell, our speculation as to the 'real' and how we choose to care are manifested in our intent. These have very real spatial manifestations, in making the spaces and places that we call destinations.

## 5. Concluding Points

Making space for intentionality and subjective meaning through highlighting our being as creative and engaging storytellers in our everyday dealings with the world recentres the human in a more-than/non-human world of the tourism encounter. Inspired by early humanistic geographers in the 1970s, the paper comes to terms with our skilful engagement with the world, here and now, whereby we negotiate proximity and boundaries with our concerns and care for the other in a broad sense. Our skills, competences and dispositions here play a role, and through the encountering of others and different ways of being and doing, we can change our own ways of being and doing. The tourism encounter is thus opened up, in one sense, by our concerned presence, enacted through the embodied habitus of care. The stories that we tell make meaning matter, whereby topos "gets close also to Pierre Bourdieu's concept of habitus" (Olsson 2007, p. 107). By implication " . . . the map is a double fold, verb turned into noun, noun to verb" (ibid, p. 115). A place is made as we make places, and in the tourism encounter, the best way to make sense of this enfolding is through the stories that we tell, exposing our intent and care.

The phenomenological enquiry can thus move tourism studies further along understanding travel as a socio-cultural material practice beyond mere business concerns, highlighting the active negotiations of proximity and boundaries, making for the places of the tourism encounter. Thereby, the phenomenology of the tourism encounter is one in which particular intentions can make explicit the role of the place itself and how it is remade in the process of relating to it in an always-unfinished way. Tourism studies which

highlight enactment, the more-than/non-human, non-representation and performance can account for this and allow for the speculative flights of fancy necessary to come to terms with our relational being, but at the same time somewhat foreclose our intentions and the forcefulness of our being. Addressing this limitation heeds the call of Pernecky and Jamal (2010) for more detailed theoretical engagement with phenomenology in tourism, grounded in realist ontology. With a focus on storytelling, the paper aligns with the call of Fendt et al. (2014) for more vivid writing and visual aids when accounting for the tourism encounter. At the same time, the paper is inspired by Crouch's (2010) 'flirting with space', allowing for our creative acumen and speculations on the real (Shaviro 2014).

At the same time I side with Giraud (2019) in stating that for all of the conceptual power invested in implicating humans in a more-than/non-human mesh and beyond, these theories can make political action impossible. What characterises the latest turnings in what has been loosely termed post-phenomenology is thus the relational ontology of the tourism encounter, which goes too far in decentring the human to meaningfully address the issues of tourism concern. Notwithstanding the aesthetics needed to account for the perpetual excess of relations and the allure of objects, I believe an explicit recognition of the limits of human intentionality and potential can be discerned, which is important in the context of the tourism encounter. In the same vein as post-phenomenology and the most recent turnings thereof, Rose et al. (2021) explored the ultimate limits of thought, simply those which cannot be known in terms of Negative Geographies, in order to

> foster more aporetic understandings of relationality, an understanding that admits the role of the nonrelational and the impossibilities immanent to relational ontologies. (see also Shaviro 2014, p. 14)

The aporetic understanding that I would like to foster is admitting the role of intent. Whilst recognising we are not alone with intent and the potential obfuscation of it through an excessive focus on the generative middle ground, we can follow Hepach (2021, quoting Carl Sauer) in renewing spatial understanding as 'chorology'. This is the study of areal differentiation, accounting for the exceptionalism of each place and the 'thick descriptions' that go with it in the Sauerian tradition, albeit wary of the 'Humboltian comprise' therein obscuring the Romantic imaginaries of place (Minca 2007). Thereby, inventorying a range of stories and ideas of place is crucial. These can set off our minds along different trajectories, helping us to envision different ways of being tourists and doing tourism. Being attentive to the here and now and to think about more than our own aspirations, and instead consider the future of life with the planet as a whole, directs our intentions to care (see Huijbens 2021b). This particular 'chorology' helps to create wayfinding maps about diversity, opportunity and potentiality. Recentring the human requires us to be plural, open and diverse, yet focused on the places and spaces that make meaning matter and how this meaning is recounted. Thereby, the challenge is to think of human intention as a specific example of a wider phenomenon, but one that matters profoundly for our common future on Earth.

The 'gentle politics' (Crouch 2010) and re-centring of the human discussed here thus revolve around that which *we* choose to care for and the stories *we* tell thereof. With focus on the dynamics of the tourism encounter, recognising more-than/non-human agency and staying truly open to the other numerous stories possible through which tourism can be conceived, the spaces and places of the tourism encounter emerge. In this way, the point is that tourism's transformative potential cannot be achieved by simply being there and going to a place. Beyond this nuance in understanding tourism 'ambassadorship' and the transformative promise of tourism, care towards the state of the planet and the lives of others in a broad sense are two obvious avenues wherein tourism matters. Stories of how we care, for what, when and how are the scholarly challenges set out by this. Addressing these can inform tourism as an act of collectivism, mutual aid and care for the other from the individual on a planetary scale.

In conclusion, I would like to evoke Olsson's (2007) figure of the border-man in his tome entitled *Abysmal*, recognising the following:

> And so it is that the limits of my world may lie less in the limits of my language and more in the limits of my imagination. (ibid, p. 247.)

Intention informs the stories that we tell, which make meaning quite literally come to matter, which is of vital importance for us, not least under the terms of the current planetary and biodiversity emergency, speculatively framed as the Anthropocene, which is one that we should all care about and can only imaginatively engage with.

**Funding:** This research received no external funding.

**Institutional Review Board Statement:** Not applicable.

**Informed Consent Statement:** Not applicable.

**Data Availability Statement:** All data are stored in accordance with the WUR data management protocol.

**Acknowledgments:** I thank the editors of the Special Issue for their interest in my work and persistence regarding the collection of essays to be published. I also thank all of the reviewers and editors for constructive and useful comments in the making of this paper.

**Conflicts of Interest:** The author declares no conflict of interest.

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
