# Peer review of "The Spaces and Places of the Tourism Encounter. On Re-Centring the Human in a More-Than/Non-Human World"

_humanities, doi:10.3390/h12040055_

Round 1

Reviewer 1 Report

Dear author and editors,

I am thankful for having the possibility to review this paper that challenges my own and my academic community's thinking about the ways to approach tourism encouters in an ontological level. I review the paper as a scholar working with tourism and travel mobilities from a posthumanist perspective, mainly grounding my work in feminist new materialisms whilst simultaneously learning constantly more about the importance of recognition of indigenous ontoepistemologies in our conceptualizations of what we've come to known as "tourism". I value highly the effort of the author to 'go against the stream', also of his own, when crafting a critique of relational ontologies' approach to tourism and in his effort to re-centralize the human. Whilst the work I'm engaged with has for the past years been to DE-centre the human from the tourism narrative I read the author's work focusing on RE-centring the human with great interest.

I had high hopes for the paper to build insights into the challenges of 'relational thinking' (in author's wording, which I would replace with 'relational ontology' that the author also uses in some cases in the paper) when it comes to practical approaches on tourism encounters, as I'm (painfully) aware of the pitfalls of understanding the world through constantly flowing and evolving relations which can make any tourism-related phenomenon, in the end, very hard to grasp and analyze. This leaves us with abstractions and theoretical statements which have very little to do with tangible realities of tourism. However, the author's need to strongly juxtapose relational ontology and phenomenology when talking about tourist encounters is confusing as the text itself reveals multiple possibilities for another kind of 'setting the scene'. Instead of starting the article with a critique of relational ontology and a somewhat stunningly old-fashioned and outdated human(mind)-centred setting, the alternative would have been to carry a post-phenomenological approach based on relations and networks of agency and a phenomenological approach based on radical humanism side by side, forming a fruitful dialogue and debate that could carry the text to its contributions to the field of tourism studies. This is all the more important given that the author approaches the tourist encounter through place and space and the meanings that are constructed for them (which is always a relational meaning-making process of which the author is, in my experience, well aware).

How could the paper have been constructed, and unfolded, if the starting point had been precisely the construction of place and the meaning of place in tourism encounters, which includes human intentionality (which I think is not denied by the post-phenomenological/relational ontological approach) and through which the author could have further argued for the importance of recognizing human agency in order to act in a time of planetary crisis? The importance of place and space in the tourist encounter is overshadowed by confrontation and incoherent argumentation, leaving the paper's contribution unfortunately weak and elusive. 

Overall, the message of the paper was unclear due to the fact that the juxtaposition presented at the beginning of the paper made it impossible to embrace the connection between the arguments in a way that would have helped the reader understand the need to re-centre the human in tourism encounter. Instead, the arguments seemed to refute each other sentence after sentence, forcing me to read the sentences over and over again, looking for a way to understand why I felt the author was arguing against himself repeatedly throughout the text. The argument provided in lines 263-266 that tells the initiative of the author to add ’an intentional body-subject to ”post-phenomenological” mix of relational emergence’, which means ’harkening back to the basics, wherein the lived spaces of being-there resurface and the human is recentred’ is one of the examples of this confusion. Whilst the human subject is according to the author is re-centred by this adding, doesn’t it only echo the necessity of human embeddedness in relations for human to exist, and in those relations the human subject having an active (intentional) role, a necessity widely acknowledged in research grounded in relational ontologies? Another argumentation in the paper, in lines 413-416, about how ”the tourism encounter needs to recognize us as spawned of the relations forged through encounters with others and the more-than/non-human”, is also an example of the confusing effect of the text. What were we arguing on behalf of, and what against? 

Being well aware of the author's previous work (as the identification of the author became possible through referencing to author's own work) which I'm highly respective for and which has strongly influenced my own work in the field, I was left in puzzlement of why this paper is not able to reach the level of argumentation that would be needed in order to make the motivation and contributions of re-centering the human clear and comprehensible.

The paper's opening with the fundamental challenge of the phenomenological inquiry seems to "box" phenomenology in "one"; leaving very little space for its nuanced nature beyond the "post-phenomenological" turn. I also find perplexing that the author refers to Shaviro's (2014) work on speculative realism in argumenting the "people centred" nature of phenomenology, instead of the author opening up the (constantly evolving) principles of phenomenology with the help of phenomenological literature. With reference to an author of a work on speculative realism, the paper immediately begins the confusing juxtaposition I mentioned earlier in the opening lines, whereby the reader no longer knows what to believe and what to not to believe.

I would really much want to see the paper to evolve to a coherent piece of argumentation, but it requires from my perspective a considerable amount of work. I would hope that the guiding thread in this work would be what the author writes in the lines 382-410; the importance of understanding human subject’s active agency as part of the unfolding relations in a more-than-human world, an agency ascribed with responsibility through ’being-there with intent’ (line 477) - an argument able to be constructed without an unnecessary juxtaposition between ”relational thinking” and ”phenomenology”.  

Other points: 

The notions of care and conviviality are mentioned in the paper some times but their elaboration is left superficial or unfinished, and I was left for longing for more understanding of how the author relates these notions to the theorization of tourism encounters through human intention, place, space, and meaning-making?

The author has constructed well the section 2. Place Ballets by opening up the grounding work of phenomenology from Husserl, Heidegger, and later Bourdieau and Lefebvre. 

I am left pondering what is the paper really asking in lines 56-59, when referring to Graber and Wengrow (2021, p. 432) in ’placing the human ”in the generative middle ground”? 

What are ’everyday utopias’ according to Cooper (2014) for a reader wanting to keep with the flow of the paper at hand and not being forced to move to a book exploring the concept?

Storytelling is mentioned in the paper but left without a deeper elaboration?

Thank you once again for the possibility to review the paper. I hope that my comments will help the author to work the paper further as I would be delighted to see it progress to a more coherent piece of text and theoretical opening.

Reviewer 2 Report

Dear Author,

the article is timely to a philosophical approach to Tourism Studies but literature in this domain is not enough explored. The link between classical and philosophical phenomenology, tourism and the purpose of building a theoretical standpoint in the research has some serious lacks. 

It would be useful to engage with some more specific international scholarship concerning phenomenology and tourism, e.g. Laura Sophia Fendt or David Crouch's writings in tourism. 

Concerning the agency of things in tourism, there are some seminal and not skippable works like those of MacCannell or Urry but also some contributions from non-representational theorists or specific by Carina Ren (in particular in Annals of Tourism Research).

Even if the article is theoretical, it would point to which areas of tourism studies are benefitting from their arguments (in conclusion, e.g.).

At this stage, the article is not publishable as it is.

Reviewer 3 Report

Dear author(s),

I am grateful for this opportunity to read and think with your thought-provoking and inspiring paper. I must admit that while I have been engaging with phenomenology and new-materialist thought in my own research inquiries, I found myself struggling to follow the argument of your paper. I agree with the argument and concern about us not recognizing the worldmaking power of our research stories and the risk of undermining ‘scope for political action’. At the same time, there is so much going on it the paper that it is challenging to read it as one, coherent piece.

After re-reading the paper, I experience that the main arguments are explained in the concluding points. This is simultaneously a philosophical and highly timely paper and I wonder whether it would somehow be made more welcoming to a wider audience; that is, to readers with less experience of phenomenological and post-phenomenological thought?

I am also hesitant or unclear about the discussions that this paper wishes to join and who it invites into the discussion. While tourism studies have been addressed clearly in the abstract and introduction, the paper does not seem to engage with phenomenological, non-representational, post-phenomenological, posthumanist, or new-materialist discussions as such. As the paper presents a concern about de-centering a human, it does not really provide references to these discussions. In other words, it would be helpful for the reader to grasp where the author(s) concerns rise. In my understanding, tourism studies are still very much human-centered, and I think there would be a need to make some kind of state of art to explain how this paper positions in those discussions. I was happy to spot the references to Doering and Zhang, but had hoped for a stronger engagement with the previous and ongoing engagements in tourism encounters – especially in connection to phenomenology. I find this somewhat contradictory to the ideas of reciprocity, responsibility, and relationality in general that the latter part of this paper advocates for. Hence, my humble suggestion is to engage with tourism studies throughout the paper in line with this Special Issue, or change the focus to reflect on the risks of more-than human/non-human turn in general.  

If you choose to recognize (and critique!) more explicitly previous research that engages with tourism encounters with phenomenological and relational thought, here are a few suggestions: Germann-Molz (2014) Camping in clearing (draws from Heidegger); Jaume Guia and Tazim Jamal (2020) A (Deleuzian) posthumanist paradigm for tourism research Tomas Pernecky, Tazim Jamal (2010) (Hermeneutic) Phenomenology in tourism studies; Höckert, Emily (2018) Negotiating Hospitality (with phenomenological thought of radical openness/hospitality); Viken, Höckert & Grimwood (2020) Cultural Sensitivity, Engaging difference in tourism, … Also the work by Intra-living in the Anthropocene-research group ilarctic.com has engaged in relational thinking with rocks and other non-humans – with new materialist thought – and perhaps their research could be included into these critical discussions as well?

As you are engaging with Heidegger’s thought, would you find the notion of hermeneutics useful here? Now the paper starts with general statements on phenomenology without recognizing the different streams within phenomenological thought. And I experience that the reader is left alone with the question of why to bring together phenomenological thought and the emerging posthumanist discussions. What is the matter of concern (borrowing from Latour) that you wish to welcome us to gather around? Without clarifying the emerging risks, as you do well in the conclusion, the reader faces difficulties in grasping the purpose of aiming to re-center the human with human-centered, non-relational approach. I am aware that this might not sound like a fair argument to make, but this was my experience and question when reading your text.

As a further suggestion: Would it also be helpful to draw attention to Karen Barad’s texts on agential realism and how those, by bringing together phenomenology, new-materialism, and quantum physics, are aiming to disrupt binary and categorical thinking between self and other (including non-human)?

And finally; would it be better to present the research question in your own words?

Apologies for the messiness of this review!

As you can read, your paper really IS thought-provoking and I am really looking forward to reading it again. All the best to the next steps!

Reviewer 4 Report

I appreciated reading the author’s engagement with post-phenomenological approaches and theories of relationality, specifically as they apply to the tourism encounter. I found the theoretical framework compelling, along with the assertion that the tourism encounter demands a theoretical approach that makes space for the role of human subjects’ meaning-making. Aside from editing notes, my main suggestion would be to establish in the paper the post-phenomenological approach to tourism that is being contested in the paper. How has a relational view been applied to the tourism encounter in problematic ways? The conclusion makes a few helpful references, but, in my view, it would make more sense, in terms of organization and development, if the application of theories of relationality to the tourism encounter preceded the author’s own argument about how that stance/view should be revised in some ways.

Here are some additional short notes:

-          The first sentence is hard to follow, particularly the second part.

-          Writing 69-70. Commas would help the reader to know how to read this sentence.

-          Writing 92-3. Punctuation.

-          Spelling 127. “Stepping.”

-          Writing 228. I think it’s just that “and” needs to be “an.”

-          Possessive 231. Geography’s?

-          Writing 237-40. I have a hard time following this sentence. Maybe use dashes.

-          Writing 383-4. Clarify the end of that sentence.

-          Line 395 – Does relational thinking really argue against distinction/difference?  

-          441 – I feel like here and elsewhere it’s important to clarify that our stories, meaning-making, etc. are one way of making matter matter – and, yes, one that’s specifically important in the tourism encounter.

-          460 – Who is saying it doesn’t? I get how we could follow a tract in philosophy that works to undermine anthropocentrism and maybe come to this conclusion, but I’m not sure that challenging human exceptionalism and our role as those who exclusively make things matter is the same as devaluing the ways we do make things matter. If that makes sense.

Round 2

Reviewer 1 Report

Thank you for the revision of the paper and the response letter where you address the changes made and reflect on your decisions in a manner that helped me in the reading experience the next version of the paper. I appreciate also your honesty in clarifying the reasons why the paper’s first version seemed undeveloped.

I was looking forward to reading the changes made in the paper as they came in such a short amount of time. The overall nature of the paper has changed rather considerably, perhaps even so much that sometimes I lost track of the grounding argument: of re-centring the human. This does not, however, mean that this is necessarily a bad thing. Now there is more space in the paper to elaborate on the topic of care as well as storytelling, which I was delighted to witness. However, I see that some of the new arguments written on the paper are somewhat rushed and end up repeating something that has already been said. In addition, some naivety is to be observed in the last sections of the paper, which I would suggest needs to be gone through in this second round of the review process. With some changes and a much needed care-full, non-rushed review of the new pieces of text by the author himself, the paper achieves in my opinion its place in being published in Humanities.

The overall flow of the paper is now better and guides the reader without leaving her in constant confusion. The strict dichotomy between relational ontology and human-centered phenomenology has now been softened, placing emphasis on intention through introducing, in more theoretically rigorous ways than in the earlier draft, the notion of care. I appreciate you taking into account de la Bellacasa’s (2017) work on speculative ethics. I could see the paper benefit from elaborating even more on care through Maria’s work? This would be interesting as her work has been one of the key drivers for feminist new materialist scholarship grounded in relational thinking. To see her work elaborated deeper on an article re-centralizing the human would be rather exciting.  

Line... 

6: “relational affect” --- affect here not needed? Does introducing affect here only make the argument unnecessarily more complex?

7: ways of doing – unnecessary and doesn’t point to anything specific 

7-8: “this paper seeks to recentre human intentionality... as the core of the tourism encounter” --- provide a short argument: why?  

8-9: “relational thinking” occurs – replace with relational ontology as you’ve done in other parts of the text? 

9-10: “augmenting with notions of care as central to understanding intention” - is this the underlying argument to be put forward in the abstract, or should it be more about augmenting with human intention, in which notions of care become of great importance? 

14: “purely alternative” - to what? 

22-27: a great addition to the introduction that prepares the reader to what’s coming 

30: “for instance animals and the material” -- non-human animals and matter? 

34: “most certainly” unnecessary 

36: “is a good thing” - rephrase to make the sentence more academically rigorous? 

37: “phenomenology has helped”... -- post-performative phenomenology?

37-38: “with a particular kind of human. The human...” --- replace end of sentence with : to continue the sentence and its logic. “...particular kind of human: the human who..:” 

(In general page 2 in introduction works much better now) 

51: “these overly relational approaches” -- rephrase? Emphasizingly relational?

52-3: approaches that want to reinstate the (post-or more-than-) human at the centreprovide examples of work that do this? (lke you do in relation to metaphysics) 

60: “this paper” -- could you here narrate yourself as “I” to enhance the flow of the text? 

73: “concomitant deliberate dismantling of collective welfare” -- open up a bit? 

76-77: very much needed argument! 

80: “this paper...” -- could you write here (and some other parts of the paper) with less repetition what that paper claims to do and write what it does without stating it doing so? 

(Multiple spelling mistakes in the text, proofreading and double checking of text by the author needed) 

Page 8: break down the very long new chapter of text (the tourism encounter).  

381: Crouch’s (2010) ‘flirting with space’ is a welcomed concept to the paper 

462: “but also to that of others” - how are them? Other people? Or more? 

485-492: can you change bullet points to flowing text? 

495: “I have proposed exploring...” -- I have earlier proposed? (to make a distinction between this paper and your earlier work) 

519-524: Really like this part. Could you have a small dialogue with Donna Haraway as you write about how it matters what stories we tell?  

529: “...as creative and engaging storytellers” -- in the context of? 

532: “...we can change” -- this argument needs connection to the tourism encounter? 

In general, in conclusion the reader would need to be reminded of the importance of re-centralizing the human earlier? (now starts in line 546) 

572: check spellings – what does “his” refer to? 

577-579: sentence very unclear – needs to be rephrased? 

581: being there & going to a place – add “being there” and “going to a place” to emphasize the notions? 

581-582: A bit naïve?? 

582: check spelling understanding  

587: check spelling   

604-607: seems as repetition and the new text in previous sections takes the “kick” out of this ending – reformulate conclusion a little bit to make it more fluent? 
